# Evidence for mirror self-recognition in beluga whales (*Delphinapterus leucas*)

Alexander Mildener◉, Diana Buchman, Sonia Ragir, Diana Reiss◉◉*

Department of Psychology, Hunter College, City University of New York, New York, New York, United States of America

◉ These authors contributed equally to this work.
* dlr28@columbia.edu

## Abstract

Tests of mirror self-recognition (MSR) have provided behavioral evidence of a high level of self-awareness in humans, chimpanzees, bonobos, orangutans, gorillas, bottlenose dolphins, Asian elephants, magpies, and to some extent in the cleaner wrasse. We conducted the standard mirror test with a social group of four beluga whales (*Delphinapterus leucas),* one subadult and three adult females, housed together at the New York Aquarium of the Wildlife Conservation Society. We exposed the whales to a two-way plexiglass mirror and a transparent control surface during baseline and post-mirror sessions and recorded and analyzed their behavioral responses in the three conditions. Two of the four whales, the subadult and her mother, exhibited a rich suite of self-directed behaviors at the mirror and subsequent mark and control sham-mark tests were conducted with both whales. The adult female showed mark-directed behavior at the mirror and passed one of the initial mark tests in a series of tests given. The self-directed behaviors exhibited by both whales and mark directed behavior by the adult female provides evidence for the capacity of MSR in the beluga whale.

## Introduction

Mirror self-recognition (MSR) reflects a form of visual self-awareness that was long considered a hallmark of human cognition. Only a handful of other species have shown the capacity for spontaneous MSR which include chimpanzees (*Pan troglodytes)* [1,2], orangutans (*Pongo abelii*) [3], bonobos (*Pan paniscus)* [4], gorillas (*Gorilla gorilla*) [5,6], bottlenose dolphins (*Tursiops truncatus*) [7–9], Asian elephants (*Elephas maximus)* [10] and Eurasian magpies (*Pica pica)* [11]. One fish species, the cleaner wrasse (*Labroides dimidiatus*) [12] has shown aspects of this ability as well.

During mirror exposure individuals demonstrating MSR generally progress through three basic behavioral stages: 1) social responses and physical exploration and

**Data availability statement:** All relevant data are within the paper and its Supporting Information file.

**Funding:** The author(s) received no specific funding for this work.

**Competing interests:** The authors have declared that no completing interests exist.

inspection of the mirror, 2) contingency testing (i.e., unusual or repetitive movements that appear to test one-to-one correspondences between one's own actions and the behavior observed in the mirror) and 3) self-directed behavior (recognition of mirror-image as self and use of the mirror as a tool to view self). Progressive changes in the stages of behavior exhibited during the mirror exposure period provide empirical evidence reflecting changes in how individuals interpret the information perceived in the mirror [1,13]. Following the emergence of self-directed behavior, individuals are tested using the standard mark test to further confirm MSR [1,14].

MSR tests with cetacean species have been limited in the number and variety of species tested due to the availability of species in aquaria that can be tested in systematic and controlled studies. Prior to the first demonstration of MSR in bottlenose dolphins [7] two studies had provided suggestive evidence for MSR in this species [15,16]. Two bottlenose dolphins demonstrated suggestive self-directed behavior during mirror exposure [15]. When exposed to self-view television that functioned as a mirror, dolphins were reported to exhibit behavior suggestive of self-examination [16]. Orcas (*Orcinus orca)* and false killer whales (*Pseudorca crassidens)* were also tested and it was reported that the animals responded with contingency testing and possible self-directed behavior suggesting MSR [17]. In contrast to studies conducted in aquaria, free-ranging Atlantic spotted dolphins (*Stenella frontalis)* were exposed to a mirror and some individuals showed exploratory and social behavior towards the mirror surface but did not display self-directed behavior [18]. The authors suggested that free-ranging dolphins may fail to demonstrate contingency testing or self-directed behavior due to greater demands on their attention made by the wild than a captive environment [18].

The beluga whale (*Delphinapterus leucas)* was a promising candidate to test for MSR. Beluga whales are members of the family Monodontidae and like the other mammalian species that demonstrate the capacity for MSR, have large complex brains [19,20] and live in complex societies with a fission-fusion social structure [21–23]. Males live in all-male bachelor pods and females and calves live in nursery pods, although mixed associations of animals of different sex and age have also been observed [23]. This species shows site fidelity and are philopatric [24,25].

Cetaceans are renowned for their hearing and echolocation ability, but their vision was long assumed to be poor. Research has shown that cetaceans do in fact possess strong visual acuity both above and below water [26], that belugas have good binocular vision [27], and that vision plays an important ecological role among cetaceans in prey capture, predator evasion and social interactions [28]. Vision in belugas may be of particular importance among cetaceans due to their unique ability to manipulate the shape of their melon for facial displays in social interactions [27,29,30]. Belugas demonstrate advanced cognitive abilities involving both their visual and auditory sensory systems across many domains similar to the capacities reported for apes, dolphins and elephants. Their cognitive prowess is evidenced by their abilities for categorical matching tasks involving vocal discrimination and call usage [31], spontaneous cross-modal

stimulus equivalence [32], relative quantity judgement [33], tool use and judgement of tool efficacy in specific tasks [34], labeling of objects through sound production [35], numerical labeling of objects [36], comprehension of mental rotation in match to sample tasks [37] and the possible use of sounds cues and communication to accomplish cooperative tasks [38] (see review [39]).

This species shows the capacity for both vocal and behavioral imitation. Belugas are vocal learners [40] and have been referred to as the *canaries of the sea* due to their proclivity for vocal imitation. Individuals have exhibited spontaneous imitation of human speech [41] and bottlenose dolphin vocalizations [42]. Belugas were trained to imitate conspecific vocalizations, synthetic computer-generated tones and the sounds of human speech [43]. They have also demonstrated the ability for behavioral imitation during play [44] and the capacity for contextual imitation of intransitive and novel body actions [45,46]. Their proclivity for the spontaneous imitation of others provides strong evidence for a high level of social awareness in this species and advanced social awareness has been linked with the emergence of mirror self-directed behavior and MSR in children and other species [2,3,47].

Over 20 years ago, we had the opportunity to expose a small social group of beluga whales housed at the New York Aquarium of the Wildlife Conservation Society to a mirror and conduct a test of MSR. Preliminary findings of the study served as a basis of an unpublished MA thesis of one of the authors (DB); however, the data was not fully analyzed at the time. In the intervening years, numerous studies conducted on beluga whale social behavior and cognition stimulated us to revisit and re-analyze the data. In this paper, we present the results of the first MSR study with beluga whales. We conducted a two-phase experiment following the standard paradigm used in previous MSR studies with dolphins [7,8]. Individuals were exposed to baseline control, mirror exposure, and post-mirror control conditions and their behavioral responses were compared across conditions. If clear self-directed behavior was observed in any of the whales, we conducted mark tests with those individuals.

## Methods

### Ethics statement

The research protocol was reviewed and verbally approved by the Animal Care Committee of the New York Aquarium of the Wildlife Conservation Society.

### Participants and facilities

We tested four beluga whales, three adult females Kathy (33 yrs), Marina (18 yrs), Natasha (21 yrs) and Natasha's female offspring Maris (7 yrs) housed at the New York Aquarium of the Wildlife Conservation Society. Marina may have had cataracts and it was unclear how this may have influenced her vision. The three adult female belugas had been captured in Churchill, Manitoba, and Maris; the offspring of Natasha, was born at the aquarium. None of the whales had exposure to mirrors prior to the onset of the experiment, however they had prior experience with semi-reflective surfaces as the windows in the pools offered differing degrees of reflectivity depending on the natural changes in light conditions. To our knowledge, none of the whales participated in prior cognitive studies.

The belugas were housed in an outdoor complex of three inter-connected above ground concrete pools, HP1, HP2 and HP2A (3.2 meters in depth). The pool complex was under a tent-like shade structure. HP1 and HP2 (each 15.0 meters in diameter) were connected by HP2A (7.5 meters in diameter). HP1 and HP2A each had two windows (148 cm high x 86 cm wide) and HP2 had a large viewing window (12.3 x 1.9 meters) for public viewing. An enclosure (211 cm x 112 cm x 278 cm) was built to surround one of the windows in HP2A. A mirror or control surface was affixed to the viewing window in the enclosure through which all video data was recorded during the experiment. The other underwater viewing window in HP2A was covered for the duration of the experiment. The whales had access to all three pools except during the last 12 sessions in which they did not have access to HP1.

## Procedures and materials

**Phase I: Baseline control, mirror exposure and post-mirror control sessions.** In Phase I the four whales were exposed to a two-way plexiglass mirror or in baseline control and post-mirror control sessions, to a transparent plexiglass control affixed to the same window. The transparent control was somewhat reflective although to a much lesser degree than the mirror despite attempts to reduce its reflective properties. The degree of reflectivity varied based on the relative light levels inside and outside of the pool (see Apparatus and Materials). Sessions were conducted in the following order: baseline control, mirror exposure, post-mirror control and a mirror exposure session.

All sessions were conducted with the whales housed together, as normally housed, to minimize stress caused by social separation. During the experiment, the mirror or control was affixed to the outside of the viewing window in HP2A. The duration of Phase I baseline control, mirror exposure and post mirror control sessions was ~ two hours. One session was conducted per day in the morning commencing between 8 and 9 am prior to the facility being open to the public. Sessions consisted of ~ one hour of *prefeed* period exposure, followed by a 5–10 minute feeding period, and then an additional ~ one hour of *postfeed* exposure. The last two sessions in Phase I consisted of ~30 minute prefeed and postfeed periods.

The feeding period between the prefeed and postfeed periods was scheduled to facilitate marking and sham-marking the belugas during Phase II. At the end of each prefeed, the trainers signaled the whales to approach an adjacent feeding area and at the end of each feed the whales were given a signal signifying the feed was over. The belugas were not touched by the trainers during the prefeed, feeding or postfeed periods during baseline control, mirror exposure or post-mirror control sessions and the mirror or control remained in place throughout the session.

**Phase II: Mirror exposure, mark and sham mark tests.** Phase II was conducted with Natasha and Maris, the two belugas that exhibited contingency testing and self-directed behavior during Phase I. In Phase II, mirror exposure sessions were conducted and interdigitated with mark and sham-mark tests. Sham-mark tests were utilized as a control to account for the tactile sensation of being marked. Sham-mark test sessions, which we refer to as *early sham-marks*, were conducted prior to the onset of mark tests. Sham-mark test sessions that were conducted after the onset of mark tests we refer to as *late sham-marks*. Mark tests were conducted in the mirror condition and early and late sham-mark tests were conducted in the mirror and control condition. During mark and sham-mark tests, the whales were marked or sham-marked by the trainers during the feed period. The procedures used in Phase II were the same as those used in Phase I except that prefeed and postfeed periods were each ~ 30 minutes in duration.

During the tests, the trainers drew a non-toxic, temporary mark or sham-mark on an area of the beluga's body that they could not see without the mirror. The shape of the marks and sham-marks consisted of a triangle, a horizontal line or a circle of ~ 5–8 cm in diameter or length. We used the same criteria for passing the mark test as used in prior cetacean studies that required the whales to orient the marked area of their body to the mirror [7–9].

**Apparatus and materials.** The two-way plexiglass mirror and the control, a transparent piece of non-glare plexiglass were the same dimensions (85 cm x 147 cm x 0.32 cm). Both the mirror and the control enabled the video documentation of the belugas' behavior in both conditions. Prior to the onset of study, examination of the mirror's reflective properties underwater by a diver and underwater photographs confirmed that the two-way mirror was an opaque reflective mirror that appeared as a normal mirror from within the pool. Viewing the mirror from inside the enclosure, it appeared as a transparent window. To optimize the reflectivity of the mirror surface facing the beluga pool, the inside of the enclosure housing the camera and mirror was kept dark during mirror conditions. The control produced minimal and differing degrees of reflectivity depending on the light level in the enclosure relative to the light level in the pool. To minimize the reflective properties of the plexiglass, the door & roof of the enclosure were opened to daylight during baseline control and post-mirror control sessions to best match the levels of light in the enclosure and pool. However, the control remained somewhat reflective and we considered it to be a less reflective but requisite control.

During baseline control and post-mirror control sessions, a photographer's blind, a curtain of matte black fabric with a hole for the camera lens, prevented the whales from seeing the camera apparatus in the enclosure and no observers were in the enclosure. Prior to the onset of sessions during the setting up of equipment, a piece of white fiberboard was used to cover the window to prevent the belugas from seeing into the enclosure. The sessions began with the two-way mirror or transparent control being affixed to the window to the pool in HP2A held in place by a 2 cm strip of black Velcro™ tape across the upper and lower edges of the window.

In the initial mark tests, we attempted to use the same type of non-toxic temporary black ink marker, (Entré, Westborough, MA) used to mark bottlenose dolphins [7], but the substance failed to adhere to the belugas' skin. Instead, we used Wet 'n' Wild black lipstick #515B (Am Cosmetics Inc., Arlington, NJ) to create a visible mark and Wet 'n' Wild Clear Gloss lipstick #554 to create a nonvisible sham-mark. The whales were sham-marked with the same brand marker used in the mark tests but filled with water. All marking and sham materials were pre-approved for topical use by the Wildlife Conservation Society animal care committee and were chosen for their waterproof qualities.

## Data collection

Data was collected by (DB) and all sessions were videotaped through the underwater viewing window in pool HP2A where either a mirror or control surface was affixed during sessions. A Sony digital8 TRV320 Handycam recorder with a Kenko 42 mm wide angle lens and a Marumi polarizer was used to record all sessions on Sony MP digital8 60-minute tapes. The camera was mounted on a tripod such that the camera lens was positioned midway between the top and bottom edge of the window.

## Data analysis

All analogue video recordings were digitized by AM using Vidbox, a video conversion program. An initial analysis was conducted by AM who reviewed and coded the behavioral responses using an ethogram (see S1 Table) during Phase I and Phase II. A subset of the videos that were comprised of contingency testing, self-directed, mark tests and sham-mark tests were reviewed and coded by DR. If there was a disparity in coding, the coders viewed the videos together and the final coding and categorization was designated if there was agreement between the two coders. Overall, the videos were analyzed in sequential order, but the coders were blind to the following conditions (i.e., baseline control, mirror exposure, post-mirror exposure control, a sub-set of the mark and sham mark tests) when coding. In a subset of the mark tests, the mark was visible that precluded a blind analysis. The duration of each digitized file varied based on the amount of information recorded over time. There were multiple files for each session and each file had a new start time.

**Phase I: Baseline control, mirror exposure and post-mirror exposure control sessions.** We quantified and compared the time spent and the number of bouts in the baseline control and the first eight mirror sessions for two of the four whales, Natasha and Maris, who exhibited the majority of time at the mirror. A bout was defined as the whales' time of arrival to time of departure from the window if it was ≥ 3 seconds and ≤ 6 ft from the window. We compared the % of time the whales spent at the window during baseline control/post-mirror control to mirror exposure sessions. We also compared the time spent in pre-and postfeed periods in the mirror and baseline control/post-mirror control sessions. The % of time was calculated by dividing time spent at the window in the different conditions by the total time of exposure.

The behavior of the four whales at the viewing window during baseline control, mirror exposure and post-mirror control sessions was coded. Behaviors exhibited while oriented to the mirror or control at a distance within ~ ≤ 6 feet were first defined as *behavioral events.* A repetitive sequence of any behavior was defined as one event. Behaviors were coded as separate events if they were discontinuous (e.g., a sequence of three firm melon presses, followed by a barrel roll, then, more firm melon presses, would be scored as two separate events of firm melon presses and one barrel roll event). All behaviors were coded using an ethogram based on previous dolphin MSR studies [7,8] (see S1 Table for ethogram and description of behaviors). The behaviors were then categorized as (1) *Social* (e.g., jaw claps), (2) *contingency testing*

including repetitive and unusual movements of the head and body (e.g., vertical head nods) (3) *self-directed* such as orienting to view parts of the body or activities otherwise inaccessible without the use of a mirror (e.g., barrel rolls) (4) stationing which included any orientation to the mirror or control in the absence of any other behavior and (5) *ambiguous behaviors* which included behaviors directed at the mirror or control that did not fall into other categories and for which the function was unclear (e.g., such as firm melon presses). All events within bouts were scored.

We quantified the frequency of occurrence of the behavioral events and categories of two of the four whales, Natasha and Maris, who exhibited significantly more bouts of longer duration and a wide variety of behavior. We also compared the frequency of occurrence of body/eye orientations when stationing in the baseline control sessions to that of the first eight mirror exposure sessions to assess if the whales showed evidence of eye preferences during mirror exposure. The codes used to categorize orientations were: head perpendicular (HPO), lateral right eye, lateral left eye, vertical right eye, vertical left eye, horizontal ventral (HVO), vertical ventral and inverted (see S1 Table for definitions of orientations). HPO, HVO, vertical ventral and inverted were all orientations involving the use of binocular vision.

**Phase II: Mark, sham-mark tests and additional mirror exposure.** We quantified and compared the % of time spent by Natasha and Maris at the mirror and control in the prefeed and postfeed periods during mark tests, sham-mark tests and additional mirror exposure sessions. All mark and sham-mark sessions conducted were analyzed to determine if the belugas oriented the marked or sham-marked area of their body to the mirror or control. Other behaviors directed towards the mirror or control were coded as the occurrence of an event but the frequency of occurrence was not calculated.

**Statistical analysis.** In Phase I Wilcoxon signed rank tests were conducted to determine if the whales that showed mirror interest 1) spent significantly more time and engaged in significantly more abouts in the mirror exposure condition compared to the baseline control, 2) spent significantly more time in the prefeed compared to the postfeed periods in the mirror and control conditions, and 3) showed a significant eye bias when stationing at the mirror. In Phase II, Wilcoxon signed rank tests were conducted to determine if the whales spent significantly more time at the mirror in the postfeed when marked compared to the prefeed when unmarked.

## Results

The study was conducted over a period of time from February through November 2001. A total of 98 sessions were conducted consisting of 27 sessions in Phase I and 71 sessions in Phase II. Due to tape degradation only 26 prefeed sessions in Phase I and 69 sessions in Phase II were included in the analysis.

### Phase I: Baseline control, mirror exposure and post-mirror exposure control sessions

In Phase I, all four whales had 15 mirror exposures sessions (~27 hours of exposure) and 12 control sessions comprised of ~23 hours of baseline control and post-mirror control sessions. Eight baseline control sessions were conducted followed by 14 mirror exposure sessions, four post-mirror control sessions and one mirror exposure session. Only two of the four whales, Natasha and Maris exhibited a wide variety of behaviors at the mirror that were categorized as social, contingency testing, self-directed, stationing or ambiguous. The other two whales stationed and showed ambiguous behaviors infrequently and rarely approached the mirror or the control during any of the sessions. Due to the insignificant length of time these two whales spent in both conditions we did not further categorize their behavior and they were not tested in Phase II.

**Time spent and number of bouts in mirror vs control sessions.** A comparison of the eight baseline control sessions with the first eight mirror sessions was conducted as the number of mirror sessions was unequal to the number of control sessions. A Wilcoxon signed-ranked test indicated that both whales spent significantly more time at the mirror than the baseline control (Natasha, $z = 2.5$, $p < .05$, Maris, $z = 2.1$, $p < .05$). Both whales also engaged in significantly more bouts in mirror sessions (Natasha, $n = 116$, Maris, $n = 128$) than in the baseline control sessions (Natasha, $n = 23$, Maris, $n = 11$) (Natasha, $z = 8.3$, $p < .05$, Maris $z = 9.4$, $p < .05$). A comparison of the % of time the whales spent overall during

Phase I mirror sessions and all control sessions (including baseline control and post-mirror control sessions) also showed Natasha and Maris spent a greater % of time at the mirror (Natasha, 3.4%, Maris, 5.4%) than at the control (Natasha 1.0%, Maris, 0.2%).

**Time spent in prefeed vs. postfeed periods in mirror and control conditions.** A comparison of the time the whales spent in prefeed and postfeed periods in the mirror ($n = 13$) and control (baseline/post-mirror exposure) conditions ($n = 11$) revealed that both whales spent more time in both conditions in the prefeed period although a Wilcoxon signed-rank test indicated the difference was only significant for Maris in the mirror condition: prefeed (56.8 min, 7.6%), postfeed (17.6 min, 2.4%), $z = -2.4$. $p < .05$. Two mirror and one control session were not included in the analysis due to missing corresponding periods.

**Behavioral events during the mirror condition.** The whales' behavioral responses to the mirror versus the baseline control and post-mirror control differed in the types, category and frequency of behaviors exhibited (see Table 1 and

Table 1. The frequency of categorized behaviors in the mirror and control conditions in Phase I.

| Behavioral Category | Behavior | Natasha # events (mirror) | Natasha # events (control) | Maris # events (mirror) | Maris # events (control) |
|---|---|---|---|---|---|
| **Social** | | **9** | **0** | **6** | **0** |
| | Jaw claps | 7 | 0 | 3 | 0 |
| | Upward head jerk | 2 | 0 | 3 | 0 |
| **Contingency Testing** | | **1** | **1** | **5** | **4** |
| | Vertical head nod | 1 | 1 | 2 | 3 |
| | Semicircular head movement | 0 | 0 | 1 | 1 |
| | Head waggle | 0 | 0 | 1 | |
| | Horizontal head shake | 0 | 0 | 1 | 0 |
| **Self-Directed** | | **35** | **4** | **68** | **0** |
| | Barrel roll | 4 | 0 | 2 | 0 |
| | Bubble bite | 12 | 3 | 34 | 0 |
| | Open mouth | 5 | 0 | 4 | 0 |
| | Pec shimmy | 0 | 0 | 4 | 0 |
| | Neck stretch | 5 | 1 | 4 | 0 |
| **CTSD (subset)** | | **9** | | **20** | |
| | Vertical head nod | 6 | 0 | 8 | 0 |
| | Semicircular head movement | 0 | 0 | 4 | 0 |
| | Head waggle | 1 | 0 | 7 | 0 |
| | Horizontal head shake | 2 | 0 | 1 | 0 |
| **Ambiguous** | | **138** | **19** | **142** | **9** |
| | Bubble production | 11 | 4 | 47 | 2 |
| | Downward slow head dip | 6 | 0 | 0 | 0 |
| | Firm melon press | 109 | 13 | 75 | 7 |
| | Melon touch | 2 | 0 | 2 | 0 |
| | Regurgitation | 0 | 0 | 1 | 0 |
| | Downward head bob | 10 | 2 | 17 | 0 |
| **Stationing** | | **119** | **44** | **102** | **10** |
| | TOTAL | 302 | 68 | 323 | 23 |

[a]CTSD refers to behaviors that first were categorized as contingency testing to be conservative but continued to occur after the emergence of self-directed behavior. These behaviors have been reported in other MSR studies of cetacean species as self-directed behaviors.

Fig 1A and 1B). In the mirror condition, Natasha and Maris exhibited a wide variety of behaviors that included social, contingency testing, self-directed, ambiguous and stationing behaviors (see Table 1, Fig 1A and 1B). The whales showed the typical progression of behavioral changes from social and contingency testing to self-directed behavior.

During the first mirror session, both whales exhibited contingency testing, ambiguous, stationing behavior and social behavior (jaw claps). No self-directed responses were observed. Both whales exhibited very few social behaviors such as jaw claps and upward head jerks (which have been described as social behavior in beluga whales [48]) towards the mirror in subsequent sessions, which typically occurred during sessions in which contingency testing or self-directed behaviors

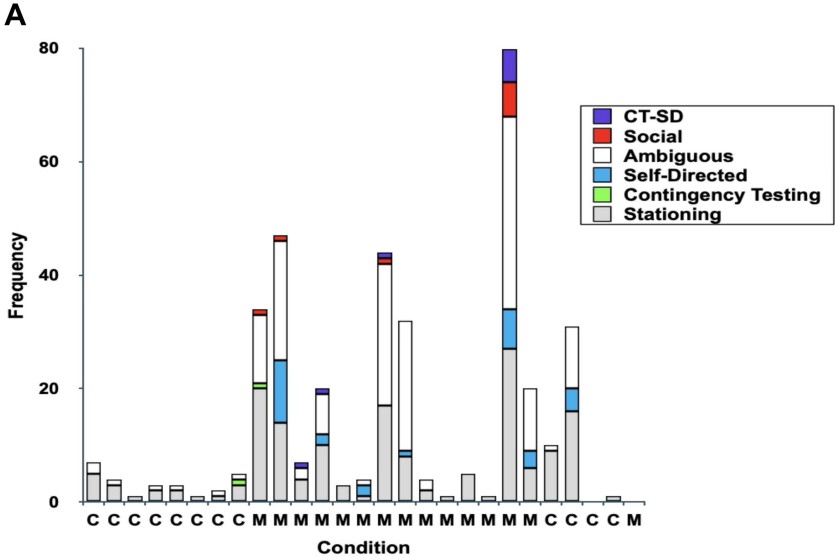

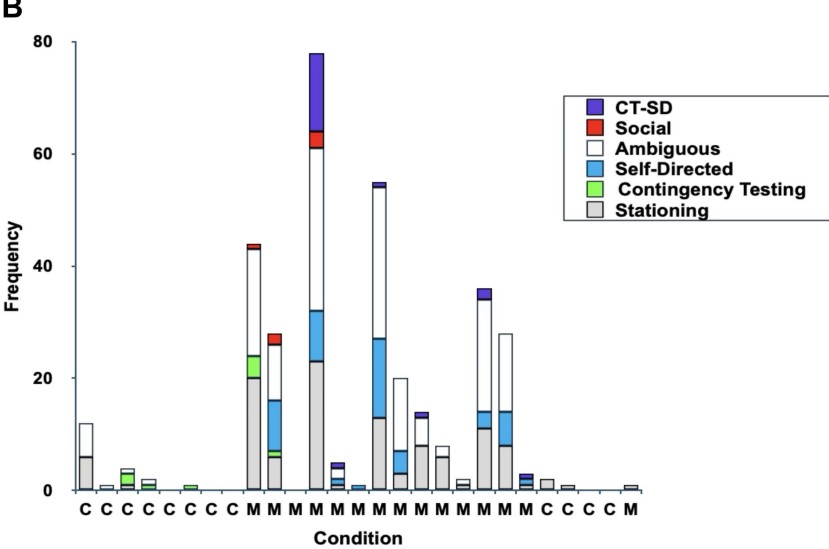

**Fig 1. Frequency of categorized behaviors for Natasha and Maris in mirror and control conditions.** (A) the frequency of occurrence of categorized behaviors exhibited by Natasha and (B) the frequency of occurrence of categorized behaviors exhibited by Maris. The session condition and order of sessions during Phase I is represented. C refers to the control condition and M refers to the mirror condition.

were also exhibited by the same whale (see Fig 1A and 1B). In these cases, it was unclear if these responses were social, contingency testing or self-directed, but to be conservative, we categorized these behaviors as social. During the first mirror session Natasha exhibited one event of contingency testing behavior, a vertical head nod and Maris exhibited a variety of contingency testing behaviors (i.e., vertical head nods, head waggles, semicircular head movements, horizontal head shakes).

Both whales engaged in a variety of what appeared to be self-directed behaviors when oriented to the mirror (Table 1, Fig 1A and 1B). Both whales exhibited these behaviors during the second mirror session after 2 hours and 11 minutes of prior mirror exposure. The first self-directed behavior by Maris was bubble bite play when alone and close to the mirror. During this event, Natasha moved behind Maris and Maris quickly departed from the mirror. Natasha then moved close to the mirror and engaged in bubble bite play. During this activity the whales produced bubbles from their blowhole or mouth and then bit the bubbles (S1 Video). Bubble bite play was the most frequent self-directed behavior to emerge in the mirror condition by both Natasha and Maris. A wide variety of other self-directed behaviors were observed throughout the mirror exposure sessions. Other self-directed behaviors included barrel rolls (S2 Video) and pec shimmies, defined as repetitive flapping of the pectoral flippers with the body oriented in a vertical plane (demonstrated by Maris only), orientations to the mirror with an open mouth and neck stretches (see S1 Table). Behaviors previously categorized as contingency testing in initial mirror sessions and baseline control sessions continued throughout later mirror sessions in which the whales were exhibiting self-directed behaviors and thus were coded as *contingency testing-self-directed behavior* (CTSD) (Table 1, Fig 1A and 1B and Discussion).

Both whales did a constellation of other behaviors at the mirror but the function was ambiguous. The predominant type of ambiguous behavior demonstrated by both whales was the firm melon press defined as pressing the melon firmly once or repeatedly against the window. It was sometimes accompanied by a stream of bubbles emitted from their blowhole. The firm melon press behavior may have functioned as contingency testing but to be conservative it was categorized as ambiguous (see Discussion). Other ambiguous behaviors included melon touches, a constellation of different types of bubble production (i.e., bubble streams, bubble bursts, bubble scants), downward head bobs, downward slow head dips (only exhibited by Natasha) and regurgitation (only exhibited by Maris). The function of these behaviors may have been contingency testing but to be conservative we categorized them as ambiguous (see Discussion).

Stationing, orienting to the mirror in the absence of other behaviors, was the most predominant response exhibited by both whales in the mirror condition (Table 1). During the first eight mirror sessions Natasha's body and eye orientations when stationing were comprised of: HPO, $n = 60$, lateral right eye, $n = 9$, lateral left eye, $n = 1$, HVO, $n = 8$ and inverted, $n = 7$. Natasha demonstrated a significant binocular eye preference compared to right eye orientations, $z = -7.0$, $p < .05$ or left eye orientations, $z = -7.5$, $p < .05$ and demonstrated a significant preference for right eye orientations over left eye orientations, $z = -2.5$, $p < .05$. Maris' body and eye orientations when stationing were: HPO, $n = 32$, lateral right eye, $n = 34$, lateral left eye, $n = 6$, HVO, $n = 14$, vertical ventral, $n = 1$ and inverted, $n = 16$. Maris also demonstrated a significant binocular eye preference compared to right eye, $z = -4.7$, $p < .05$ or left eye orientations, $z = -6.6$, $p < .05$ and exhibited a significant preference for right eye over left eye orientations, $z = -4.6$, $p < .05$.

**Behavioral events during the baseline and post-mirror control conditions.** During the baseline control sessions, only contingency testing, ambiguous and stationing behavior was demonstrated by the belugas. Both whales exhibited what appeared to be contingency testing behaviors, sequences of vertical head nods, head waggles and semicircular head movements. These behaviors were first observed by Maris in the third and by Natasha in the final baseline control session. A few instances of these behaviors were exhibited infrequently in the other baseline control sessions for Maris. No self-directed behaviors were observed by either whale in the baseline control sessions.

Natasha exhibited four instances of self-directed behavior which all occurred in one post- mirror control session. This session was after Natasha had prior mirror exposure sessions ($n = 14$) in which she exhibited self-directed behaviors.

In this one control session, she exhibited three events of bubble bite play and a neck stretch, behaviors she previously exhibited towards the mirror (see Discussion). Maris did not demonstrate any self-directed behavior toward the control.

The whales showed ambiguous behaviors during the baseline control and/or post-mirror control sessions. The predominant type of ambiguous behavior demonstrated by both whales was the firm melon press. Other ambiguous behaviors included downward head bobs (only Natasha) and a constellation of different types of bubble production (i.e., bubble streams, bubble bursts, bubble scants). Natasha exhibited ambiguous behaviors in the baseline control ($n=7$) and in post-mirror control sessions ($n=12$). Maris only demonstrated ambiguous behavior in the baseline control sessions ($n=9$).

The predominant behavior of both whales in all conditions was stationing (Table 1) (see Fig 1A and 1B) Natasha: baseline control sessions ($n=18$); Maris ($n=7$) and post-mirror control sessions Natasha ($n=26$); Maris ($n=3$). Stationing was the only category of behavior demonstrated by Maris in the post-mirror control sessions (see Fig 1A and 1B).

Natasha's orientations in the baseline control when stationing were comprised of: HPO, $n=16$, lateral right eye, $n=3$ and lateral left eye, $n=1$. When stationing Natasha demonstrated a significant binocular eye preference compared to right eye orientations $z=-3.2$, $p<.05$ or left eye orientations, $z=-3.4$, $p<.05$. The type and frequency of occurrence for Maris' orientations in the baseline control included HPO $n=3$, lateral right eye, $n=1$, HVO, $n=1$ and inverted, $n=2$. There were such few instances of left eye and right eye orientations in the baseline control, no statistical analysis could be conducted to compare right to left eye viewing. Maris demonstrated such few instances of stationing in the baseline control that no statistical analysis could be conducted.

### Phase II: Mirror exposure, mark tests, sham-mark tests

Two of the four whales, Natasha and Maris, demonstrated a variety of self-directed behaviors in Phase I and were further tested in Phase II in which we conducted sham-mark tests, mark tests and additional mirror exposure sessions. A series of early sham-mark tests and four mirror exposure sessions were conducted prior to the onset of mark tests with both whales. After the mark tests commenced, late sham-mark tests and two additional mirror exposure sessions were interdigitated with the mark tests (see Fig 2 for diagram of mark and sham-mark locations on their bodies, see S2 Table for mark and sham-mark location and order of testing).

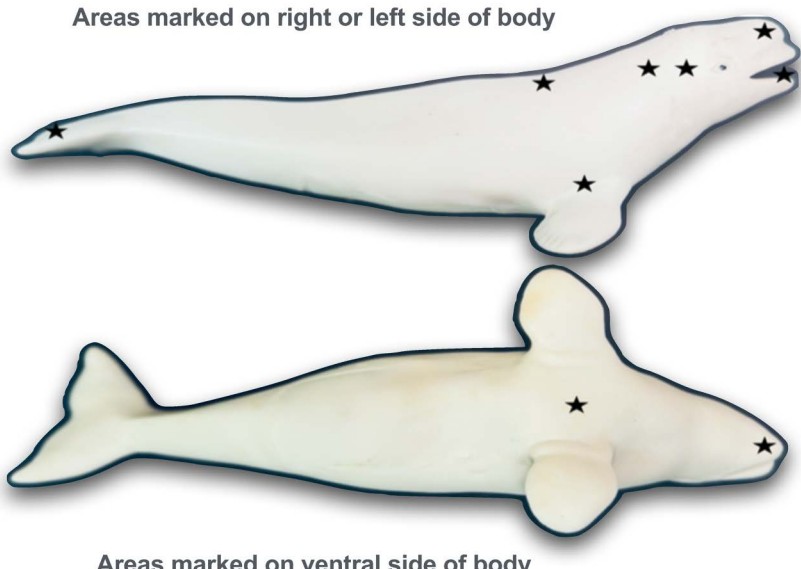

**Areas marked on right or left side of body**

**Areas marked on ventral side of body**

**Fig 2. The locations of marks and sham-marks on Natasha and Maris during mirror and control tests.**

**Early sham-mark tests.** Early sham-mark control tests were conducted in the mirror condition (Nat, $n=15$, Mar, $n=6$) and non-reflective control condition (Nat, $n=9$, Mar, $n=3$) with Natasha and Maris. Due to degradation of the tapes, only two of the three videotaped sessions of Maris in the control condition could be digitized and analyzed. Neither Natasha or Maris oriented the sham-marked areas of their body toward the mirror or control.

Both whales spent a greater % of time at the mirror in the prefeed (Nat, $n=7.0\%$, Mar, $n=1.7\%$) than the postfeed (Nat, $n=4.1\%$, Mar, $n=.07\%$). Natasha spent more time at the non-reflective plexiglass control in the postfeed, $n=4.0\%$ than prefeed, $n=2.6\%$, while Maris spent more time during the prefeed, $n=0.3\%$ than postfeed, $n=0\%$.

**Phase II mirror exposure.** Six mirror exposure sessions were conducted. Natasha spent a greater % of time at the mirror in the postfeed, $n=3.1\%$ than prefeed, $n=1.6\%$, while Maris spent a greater % of time in the prefeed, $n=6.9\%$ than postfeed, $n=4.7\%$ and only approached the mirror during five of the six sessions.

**Mark test: Natasha.** A total of 14 mark tests were conducted with Natasha, but only 13 mark tests were included in the analysis as one session could not be digitized due to tape degradation. During the mark tests Natasha spent time at the mirror in eight prefeed periods and in the first three postfeed periods (MT1, MT2, MT3) (see Fig 3). Natasha spent a greater % of time at the mirror in the postfeed when marked ($n=1.3\%$) as compared to the prefeed when unmarked ($n=1.2\%$). A Wilcoxon signed rank test indicated that the difference was not statistically significant $z=-0.7$, $p>.05$.

In the first mark test (MT1) Natasha engaged in a greater number of bouts and spent more time in the postfeed when marked ($n=4$ bouts, 49 s) as compared to the prefeed when unmarked ($n=1$ bout, 3 s). No statistical analysis could be conducted to compare time spent during the prefeed compared to the postfeed due to the low number of bouts exhibited. Natasha was marked on the right of her melon and only oriented with her head perpendicular to the mirror (HPO). During two bouts she exhibited a downward slow head dip. Although the mark was visible during HPO orientations, she did not

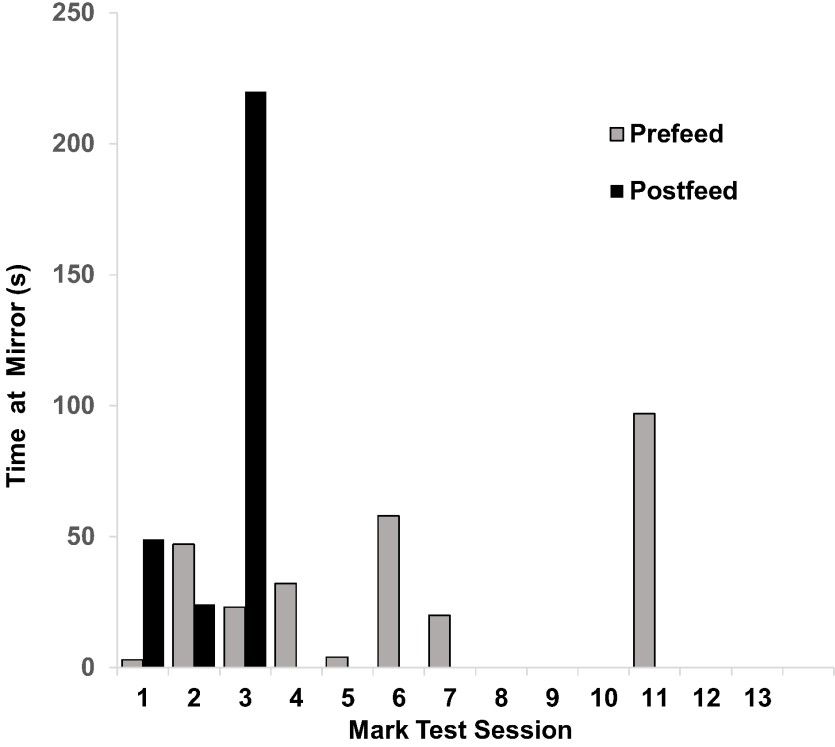

**Fig 3. Natasha Mark Test: Time spent at mirror during mark tests in prefeed vs postfeed periods when marked.** Natasha passed the third mark test by orienting the marked area on her body to the mirror repeatedly and spent the greatest amount of time in the postfeed period.

specifically orient the marked area of her body to the mirror. We did not consider this orientation sufficient for passing this mark test. In her second mark test, Natasha engaged in a greater number of bouts and spent more time at the mirror in the prefeed when unmarked ($n=4$ bouts, 47 s) compared to the postfeed when marked ($n=2$ bouts, 24 s). No statistical analysis could be conducted to compare time spent during the prefeed compared to the postfeed due to the low number of bouts exhibited. Natasha was marked on the left side of her melon and she demonstrated a lateral left and HPO orientation during one bout. However, a single bout in a left eye orientation was not considered sufficient for passing this mark test.

Natasha passed her third mark test (MT3). Natasha did a greater number of bouts and spent more time in the postfeed when marked ($n=14$ bouts, 220 s/3.7 min) compared to the prefeed when unmarked ($n=4$ bouts, 23 s). A Wilcoxon signed rank test indicated that she engaged in a significantly greater number of bouts and spent significantly more time in the postfeed when marked compared to the prefeed when unmarked bouts: $z=2.8$, $p<.05$, time: $z=3.1$, $p<.05$. Natasha was marked behind her right ear during this mark test. In the postfeed, prior to her first mark-directed orientation to the mirror, she did a sequence of repetitive laps ($n=9$) around the pool for 1.6 minutes and pressed the marked area of her body against the mirror in 6 of the 9 laps. She continued to swim freely around the pool and she oriented the marked area of her body in 12 of the 14 bouts. During her mark-directed orientations, Natasha positioned vertically and horizontally with the right-marked side of her body towards the mirror. She oriented the marked area of her body to the mirror more often in the postfeed period (187 s/220 s, 85%) as compared to the prefeed period (8 s/23 s, 35%).

Natasha's behavior when marked during this session was also distinctive from her behavior in all other sessions. In this postfeed she engaged in her only instance of toy manipulation at the mirror. She swam away from the mirror to a hoop at the opposite side of the pool, returned with it to the mirror, released the hoop while orienting the marked area of her head towards the mirror. During this event she did a downward head bob simultaneously opening and closing her mouth and then took the hoop in her mouth and dove out of view to the bottom of the pool (S3 Video). In this postfeed session she also displayed a suite of self-directed and ambiguous behaviors which included sequences of repetitive downward head bobs, an instance of rapid and vigorous horizontal head shaking, opening and closing her mouth and five instances of downward head bobs (S4 Video). This was the only session where she exhibited a repetitive sequence of downward head bobs.

**Mark test: Maris.** Maris was marked in 15 sessions and only spent time at the mirror in six of the prefeed periods and in three of the postfeed periods. She spent more time at the mirror in the prefeeds than the postfeeds across all of her mark tests except in MT5. A Wilcoxon signed rank test demonstrated that Maris showed no significant difference in time spent at the mirror in the prefeed (126 s/2.1 min, 0.5%) compared to the postfeed (30 s, 0.1%), $z=-1.6$, $p>.05$.

In MT5 Maris was marked on the right side of her head, however, the mark was not visible in the video. It is possible the mark degraded or was brushed off by Maris. Notably she oriented to the area where the mark had been applied, but because no mark could be seen, we did consider this a successful passing of the mark test.

**Late sham mark tests.** Late sham-mark tests were conducted only with Natasha who showed mark-directed behavior. Three late sham-marks, two in the mirror condition and one in the control condition, were interdigitated with the mark tests. Natasha did not orient the sham-marked area of her body to the mirror or control. Natasha spent more time at the window in the postfeed in both conditions (prefeed mirror: 0.6%, postfeed mirror: 1.1%, prefeed control: 0.6%, postfeed control: 2.6%).

**Self-directed behaviors during phase II.** Several of the self-directed behaviors and CTSD behaviors exhibited at the mirror by the whales in Phase I continued in Phase II and were observed during mirror exposure sessions, sham-mark tests and mark tests. Both whales exhibited other new self-directed behaviors at the mirror during this phase that included repetitive up-down swims and toy manipulation. During up-down swim events the whales would approach or be stationed at the mirror and do repetitive sequences of rising upwards and downwards from the top to bottom edges of the mirror. Toy manipulations at the mirror were observed by both whales. Natasha's only occurrence of toy manipulation was during

the mark test she passed (MT3). Maris engaged in a total of 14 sequences of toy manipulation at the mirror that occurred during two sessions. In one session, she brought a knotted rope toy to the mirror, did a barrel roll while holding it in her mouth, then approached the mirror more closely, moved her body in a slow and circular motion, and released the toy. She then dove down, brought the toy back to the mirror, did several other orientations while holding the toy in her mouth, did a firm melon press and then swam away from the mirror (S5 Video). Maris did other sequences of toy manipulation at the mirror in these two sessions in which she held the rope toy in her mouth often while doing other concurrent behavior including firm melon presses, barrel rolls, bubble streams and shifting her eye-body orientations.

## Discussion

Phase I of the study exposed four beluga whales to a mirror and a semi-reflective control providing them with the opportunity to experience the contingencies of mirror use. The whales showed different degrees of interest in the mirror and control as measured by the number of bouts and time spent in the two conditions. Two of the whales, Marina and Kathy spent little time at the mirror and when they did so, failed to demonstrate social, contingency testing or self-directed behavior. The other two whales, Natasha and Maris, spent more time at the mirror as compared to the control as evidenced by their percentage of time spent and number of bouts in each condition. Both Natasha and Maris demonstrated the same behavioral progression from social and contingency testing to self-directed behavior that has been a hallmark of the behavior of animals that have demonstrated MSR. This change in behavior provides empirical evidence of their changing interpretation of the 'other' in the mirror.

Natasha passed the third mark test (MT3) by orienting the marked area of her body, the area behind her right ear, towards the mirror while exhibiting a rich suite of self-directed behaviors as previously described in the results section. She also spent significantly more time at the mirror when marked in the postfeed compared to when unmarked in the prefeed. The passing of a mark test by Natasha provides further evidence for this capacity in a beluga whale. In the first mark test (MT1) Natasha was marked on the right side of her melon and although she did not orient this area towards the mirror, she spent more time at the mirror in the postfeed when marked compared to the prefeed. Why Natasha failed to attend to the marks placed on her body in other mark test sessions remains unknown. Some chimpanzees who typically pass their first mark test, have been reported to habituate and lose interest in subsequent mark tests [49]. Maris did not pass any of her mark-tests.

Although Maris did not pass any of the mark tests, the variety of self-directed behaviors she exhibited at the mirror provides suggestive evidence of her capacity for MSR. It has been suggested that MSR might be effectively demonstrated by self-directed behavior in lieu of the classic mark litmus test especially in animals without prehensile appendages that allow them to touch the mark [7,8,50]. In 1994, Gallup was the first to suggest that the emergence and demonstration of self-directed behavior was in itself evidence of MSR but devised the mark test to further confirm this capacity [51]. Since then, it has been proposed that self-directed behavior independent of the mark test should be considered as sufficient evidence of MSR [8,52].

### Progressive stages of behavior

There was a paucity of social behaviors observed throughout the study and this may be due in part to prior exposure the whales may have had with reflective surfaces of their pool windows. As in the initial mirror exposure phase of MSR studies conducted with other cetaceans (bottlenose dolphins, orcas, and false killer whales) the belugas in this study did not display social behaviors as reported in studies with ape species. It has been suggested that this might be due to their prior experience with reflective surfaces in the aquarium environment [7,8]. Bottlenose dolphins in managed care that lacked experience with reflective surfaces initially responded to a mirror with social behavior [53]. A study with free-ranging Atlantic spotted dolphins, reported one instance of social behavior in the wild, with most individuals ignoring the mirror [18].

Notably, in the later sessions of Phase I, Natasha and Maris exhibited upward head jerks and jaw claps, behaviors generally categorized as social agonistic behaviors [8,48,54] during sessions in which they also exhibited contingency testing and self-directed behaviors (e.g., bubble bite play). Apes [2] and dolphins [8] also exhibited similar occurrences of presumably agonistic "social behaviors" toward the mirror after the onset and concurrent emergence with self-directed behaviors. The authors of both studies suggested that such instances of what looked like 'social gestures' along with solitary motor and object play were best explained as self-monitoring and self-directed [2,8].

The behaviors we initially categorized as contingency testing behaviors included vertical head nods, semicircular head movements, head waggles and horizontal head shakes were similar to behaviors reported in other MSR studies with bottlenose dolphins [7,8] and orcas [17]. These behaviors initially categorized as contingency testing may in fact have been self-directed behaviors as they continued to occur after the onset of self-directed bubble bite play at the mirror. The demarcation and distinction between contingency testing and self-directed behavior is often difficult as the function of the behaviors may change over time [7,8]. Initially, these unusual repetitive behaviors may function as contingency testing, providing individuals with the opportunity to test the contingencies of their behavior at the mirror much like when we wave in a store security monitor to determine if we are seeing ourselves. This stage of contingency testing is followed by the emergence of self-directed behavior and behaviors previously categorized as contingency testing that continue to be exhibited may now function as self-directed behaviors. For this reason, we use the term CTSD to represent this change in possible function.

Natasha and Maris engaged in a rich suite of self-directed behaviors that emerged early in both whales in the second mirror session. Bottlenose dolphins also demonstrated the early emergence of self-directed behavior during their first mirror or fourth mirror session [8]. The first self-directed behavior at the mirror observed in both belugas was bubble bite play- an activity in which the belugas emitted bubbles from their mouth or blowhole and then bit the bubbles as they were produced. The whales appeared to be using the mirror as a tool to observe themselves engaging in this activity. Bubble biting is a form of play in belugas that has been documented in the literature. The production and manipulation of clouds, streams and single bubble rings is a form of solitary play common among belugas in captivity [44,48,55,56]. In MSR tests of dolphins it was reported that individuals at the mirror produced and played with bubbles and these events were categorized as self-directed behaviors [8]. Other self-directed behaviors exhibited by Natasha and Maris at the mirror included barrel rolls, pec shimmies, vertical ventral neck stretches and open mouths. Barrel rolls and neck stretches were also reported as self-directed behaviors observed in bottlenose dolphins [8]. Natasha and Maris both oriented to the mirror with open mouths and this open mouth behavior at the mirror was observed in MSR studies with bottlenose dolphins, orcas and false killer whales [7,8,17].

During Phase II, both belugas engaged in another form of self-directed behavior, toy play and manipulation. As previously described in the results, during the mark test that Natasha passed, after orienting to the mark she exhibited her only occurrence of toy manipulation, bringing a toy to the mirror and interacting with it. Maris also engaged in complex sequences of self-directed behaviors involving toy manipulation and play at the mirror such as barrel rolling while holding a knotted rope toy in her mouth as well as the other toy interactions at the mirror previously described in the results. Such self-directed behaviors provide compelling evidence in support of the belugas' capacity for MSR. In other tests of MSR, other species, bottlenose dolphins [7], orcas [17] and elephants [10] have been observed to transport objects or food to a mirror.

During baseline control sessions prior to mirror exposure, Natasha and Maris exhibited contingency testing behavior toward the control. Natasha also exhibited self-directed behavior toward the control in one post-mirror control session. Notably, she never exhibited self-directed behavior during any of the baseline control sessions prior to mirror exposure. These events of self-directed behavior were all during one session conducted after eight prior mirror sessions in which she exhibited suites of self-directed or CTSD behaviors. We hypothesize that Natasha's prior use of the mirror to view herself as evidenced by her self-directed behavior, may have contributed to the saliency and function of the control as a reflective surface. Furthermore, as stated in the methods section, the reflective properties of the control varied depending

on the light level in the research enclosure relative to the light in the pool. It is possible that the control was more reflective in that particular session. Maris only demonstrated self-directed behavior at the mirror.

**The ambiguity of ambiguous behaviors.** We assigned the category ambiguous to those behaviors for which the function was unclear. These behaviors included firm melon presses, melon touches, downward slow head dips, regurgitation, downward head bobs and different forms of bubble production. The firm melon press was the second most predominant behavior (stationing was the most predominant behavior) for both Natasha and Maris. We hypothesize that it may have functioned as contingency testing or CTSD behavior functioning to inform the whales about the visio-tactile properties of the mirror and/or afford the whales the opportunity to view and test their changing image as they approached, touched and receded from the mirror. It could also allow them to test and confirm if they were seeing their own image. Echolocation directed at the mirror image would not provide such information about an individual and its reflection. As discussed above, behaviors initially categorized as contingency testing may later function as self-directed behaviors when observed concurrently with other self-directed behaviors. As the function of firm melon presses used in this context was unclear, we categorized it as ambiguous.

Other behaviors such as bubble production (i.e., bubble streams, bubble bursts and scants), which sometimes were exhibited concurrently with firm melon presses, occurred less frequently. Bubble production at the mirror was coded as ambiguous but may have been self-directed behavior. Another behavior categorized as ambiguous was the single downward head bob which was often observed in both belugas as they swam by the mirror. These single events were not included in the analysis because they occurred outside mirror bouts but may have functioned as contingency testing or social behavior although this behavior has not been previously reported as a social behavior in belugas. As the function was unclear it was coded as ambiguous. Regurgitation, downward slow head dips and melon touches were coded as ambiguous due to the uncertainty of their function.

Stationing was the predominant behavior demonstrated towards the mirror and control for both belugas. It is possible that stationing could be a form of self-directed behavior as the belugas may have been using the mirror as a tool to watch themselves in their various orientations, however this remains unclear. Stationing was assigned its own category to be conservative. When stationing, both Natasha and Maris demonstrated a significant binocular eye preference but right eye viewing was significantly higher than left eye viewing at the mirror. The belugas' preference for binocular viewing when at the mirror is consistent with reports of belugas demonstrating a binocular eye preference when viewing objects and people [57,58]. It has been reported that the thicker and more anteriorly oriented frontal groves/optic canals in the beluga eye suggest good binocular vision [27]. It has also been reported that belugas show a left eye over right eye preference when inspecting objects [58] which differs from our findings of a right eye preference when viewing self. There is evidence that belugas may demonstrate a left eye preference when viewing conspecifics. Behavioral observations of belugas in the wild and in human care have reported that the calves determine their spatial placement relative to their mother and that calves typically place themselves to the right of their mother thus maintaining their left eye bias [59–61]. Other cetaceans such as bottlenose dolphins have also demonstrated a left eye bias when viewing conspecifics [62]. The fact that Natasha and Maris rarely demonstrated left eye viewing could suggest that they did not see their mirror image as a conspecific. However, further studies on visual laterality in belugas should be conducted.

## Limitations

There were a few limitations in this study. It would have been preferrable to employ a totally non-reflective control rather than a semi-reflective control. We opted to use the latter as it afforded the opportunity to videotape and compare the intricacies of the whales' behavioral responses to the mirror.

## Conclusion

This study provides evidence for the capacity for MSR in another cetacean species, the beluga whale. The emergence of self-directed behavior in two beluga whales and mark-directed behavior in one beluga whale after exposure to a mirror

indicates that this capacity may be more widespread beyond the family Delphinidae and extend to Monodontidae. Future studies with additional beluga whales of different ages and sexes are needed to advance our understanding of this capacity in this species.

## Supporting information

**S1 Table. Ethogram of orientations and behaviors exhibited at the mirror or control.**
(PDF)

**S2 Table. Mark and sham-mark location and order of testing.**
(PDF)

**S1 Video. Natasha bubble bite.** Natasha demonstrating self-directed behavior via bubble bite play.
(MP4)

**S2 Video. Maris Barrel roll.** Maris demonstrating self-directed behavior via a barrel roll.
(MP4)

**S3 Video. Natasha mark test toy.** Natasha demonstrating mark-directed behavior and toy manipulation.
(MP4)

**S4 Video. Natasha mark test.** Natasha demonstrating mark-directed behavior.
(MP4)

**S5 Video. Maris toy manipulation.** Maris engaging in toy manipulation and other self-directed behaviors.
(MP4)

**S1 Data. MSR data.**
(XLSX)

## Acknowledgments

We thank Dr. Heidi Lyn and Dr. Margaret Stanton for their assistance in data collection and their preliminary analysis of the data and Dr. Martin Chodorow for his guidance and suggestions concerning statistical analysis. We also express our deep gratitude to the beluga whale animal care team at the New York Aquarium of the Wildlife Conservation Society for their participation and assistance in this study and the Wildlife Conservation Society for their in-kind support of this research. We dedicate this manuscript to Hvaldimir, the beluga whale who fostered an appreciation for cetaceans all over the world.

## Author contributions

**Conceptualization:** Diana Reiss, Diana Buchman.

**Data curation:** Diana Reiss, Alexander Mildener, Diana Buchman.

**Formal analysis:** Diana Reiss, Alexander Mildener.

**Investigation:** Diana Reiss, Alexander Mildener, Diana Buchman.

**Methodology:** Diana Reiss, Alexander Mildener, Diana Buchman.

**Project administration:** Diana Reiss.

**Resources:** Diana Reiss.

**Supervision:** Diana Reiss.

**Validation:** Diana Reiss.

**Visualization:** Diana Reiss, Alexander Mildener.

**Writing – original draft:** Diana Reiss, Alexander Mildener, Sonia Ragir.

**Writing – review & editing:** Diana Reiss, Alexander Mildener, Diana Buchman, Sonia Ragir.

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
