## [Decision Letter · Decision Letter 0]

2 Mar 2026

PONE-D-26-01940Evidence for mirror self-recognition in beluga whales (Delphinapterus leucas)PLOS One

Dear Dr. Reiss,

Thank you for submitting your manuscript to PLOS ONE. After careful consideration, we feel that it has merit but does not fully meet PLOS ONE’s publication criteria as it currently stands. Therefore, we invite you to submit a revised version of the manuscript that addresses the points raised during the review process.

Both reviewers agree that your study adds valuable data to the subject of MSR in vertebrates, particularly considering the challenges of working with cetaceans. However, reviewers also raise important questions and concerns that need to be addressed.

If applicable, we recommend that you deposit your laboratory protocols in protocols.io to enhance the reproducibility of your results. Protocols.io assigns your protocol its own identifier (DOI) so that it can be cited independently in the future. For instructions see: https://journals.plos.org/plosone/s/submission-guidelines#loc-laboratory-protocols. Additionally, PLOS ONE offers an option for publishing peer-reviewed Lab Protocol articles, which describe protocols hosted on protocols.io. Read more information on sharing protocols at . Additionally, PLOS ONE offers an option for publishing peer-reviewed Lab Protocol articles, which describe protocols hosted on protocols.io. Read more information on sharing protocols at https://plos.org/protocols?utm_medium=editorial-email&utm_source=authorletters&utm_campaign=protocols..

We look forward to receiving your revised manuscript.

Kind regards,

Ulrike Gertrud Munderloh, Ph.D.

Academic Editor

PLOS One

Journal Requirements:

Reviewers' comments:

Reviewer's Responses to Questions

**Comments to the Author**

1. Is the manuscript technically sound, and do the data support the conclusions?

Reviewer #1: Yes

Reviewer #2: Yes

2. Has the statistical analysis been performed appropriately and rigorously? 

Reviewer #1: Yes

Reviewer #2: Yes

3. Have the authors made all data underlying the findings in their manuscript fully available?

Reviewer #1: Yes

Reviewer #2: Yes

4. Is the manuscript presented in an intelligible fashion and written in standard English?

Reviewer #1: Yes

Reviewer #2: Yes

5. Review Comments to the Author

Reviewer #1: I consider this paper to be a valuable contribution to MSR research in a cetacean species for which experimental opportunities are limited. I was pleasantly surprised to learn that the experimental video material dates back to 2001. Although many of the results are descriptive, the data set is clearly worthy of publication, and I appreciate the authors’ efforts in compiling and analyzing these materials.

Among the four individuals, two showed clear interest in the mirror. However, given that long-tem captive belugas may have previously encountered reflective surfaces such as tank glass, it is quite possible that they already had some degree of mirror-like experience. In addition, considering individual differences, the absence of interest in some is not unexpected.

Although only Natasha passed the mark test, and only in a single trial, the self-directed behaviors observed in Natasha and Moris provide strong evidence of MSR, I think. Overall, I find the conclusions of this paper to be reasonable.

In recent MSR research, the focus has increasingly shifted from whether a given species is capable of MSR to how MSR is acquired. For example, a mark test study using 9 fish of the cleaner wrasse demonstrated that MSR was established as early as 20 minutes in the fastest individuals, and on average about 80 min after mirror exposure (Sogawa et al. 2025, Scientific Reports). Prior to the establishment of their MSR, cleaner wrasse exhibited social behaviors (aggression to own mirror image) and contingency testing (C-testing) behaviors. These findings suggest that behaviors previously interpreted as C-testing may occur after establishing MSR, and may therefore reflect examination of the properties of the mirror rather than the own mirror image. Similar interpretations have been proposed in many MSR studies, including those on chimpanzees (Chimps will be within 30 min, Povinelli et al. 1993).

From this perspective, it would be interesting to consider how long after initial mirror exposure belugas may be able to establish MSR. If possible, I hope the authors to discuss this point in light of the present findings. In Sogawa et al. (2025), individuals of cleaner wrasse were observed to lifting food items (pieces of shrimp) near the water surface in front of the mirror and dropping them to observed their movement. This behavior occurred after establishment of MSR, and is interpreted as an examination of the mirror’s properties. Distinguishing between behaviors directed at the self-image and those directed at the mirror itself may provide additional insight into the interpretation of mirror-related behaviors.

Finally, I note that the citation style in the reference list was not entirely consistent, please make them in the form.

Reviewer #2: Consider the role of ecology when it comes to mark saliency in Beluga whales. Beluga ecology does not rely on vision as a primary sensory modality the way primates and birds do. Ecological context may be useful when comparing cetacean species tested, in particular to bottlenose dolphins where there is evidence of capacity. Kohda et al (2012) demonstrated the importance of a species ecological context with cleaner wrasse who are biologically wired to attend to unusual visual marks.

Consider the weakness of using a mark test relying on vision in a species where there may not be an ecologically relevant reason for them to attend to the mark. This is where tests for mark saliency such as in Prior et al. (2008) in magpies become useful to demonstrate the species show interest in the mark.

Consider referencing papers that have explored mark tests in alternative modalities to address this possibility (eg the olfactory test developed by Cazzolla Gatti (2016) for dogs).

In the limitations the author’s state;

“It would have been preferrable to employ a totally non-reflective control rather than a semi-reflective control.”

It may be worth recontextualising the role of reflective surfaces when it comes to captive aquatic animals, as their exposure and experience to reflective surfaces may be a key detail impacting the promising results in captive species. Studies where animals are given the mark test after a set number of hours without the prerequisite emergent self-directed behaviours often fail. Too much exposure to reflective surfaces means that the once novel properties may no longer elicit exploratory or curious behaviours.

A lack of social behaviours towards a reflection could indicate understanding of mirror properties, but if they do not display self-directed or mark directed behaviours, their understanding cannot be quantified as easily.

It is also important to situate these findings within ongoing methodological debates and differences in opinion in MSR literature. Different MSR researchers often have varying thresholds for evidence of MSR capacity in a species, often limiting evidence to great apes, and select corvids. Lack of replication within an individual, and between individuals is typical for MSR research, and requires detailed methods as seen in this paper so that the full context of the individual animals tested can be understood by readers. In this case, while replication was limited within and between individuals, the response occurred under controlled conditions, with careful and justified behavioural categorisations strengthening its interpretive weight.

6. PLOS authors have the option to publish the peer review history of their article (what does this mean?). If published, this will include your full peer review and any attached files.). If published, this will include your full peer review and any attached files.

.

Reviewer #1: No

Reviewer #2: **Yes:** Eva KakradaEva Kakrada

---

## [Author Response · Author response to Decision Letter 1]

11 Apr 2026

Please see the attached document, Response to Reviewers. We deeply appreciate the comments and suggestions provided by the editor and two reviewers and have addressed the points raised.

---

## [Editor Report · Decision Letter 1]

14 Apr 2026

Evidence for mirror self-recognition in beluga whales (Delphinapterus leucas)

PONE-D-26-01940R1

Dear Dr. Reiss,

We’re pleased to inform you that your manuscript has been judged scientifically suitable for publication and will be formally accepted for publication once it meets all outstanding technical requirements.

An invoice will be generated when your article is formally accepted. Please note, if your institution has a publishing partnership with PLOS and your article meets the relevant criteria, all or part of your publication costs will be covered. Please make sure your user information is up-to-date by logging into Editorial Manager at Editorial Manager® and clicking the ‘Update My Information' link at the top of the page. For questions related to billing, please contact  and clicking the ‘Update My Information' link at the top of the page. For questions related to billing, please contact billing support..

Kind regards,

Ulrike Gertrud Munderloh, Ph.D.

Academic Editor

PLOS One
---

## [Editor Report · Acceptance letter]

PONE-D-26-01940R1

PLOS One

Dear Dr. Reiss,

I'm pleased to inform you that your manuscript has been deemed suitable for publication in PLOS One. Congratulations! Your manuscript is now being handed over to our production team.

Kind regards,

on behalf of

Dr. Ulrike Gertrud Munderloh

Academic Editor

PLOS One